# Asymmetric Push–Pull Type Co(II) Porphyrin for Enhanced Electrocatalytic CO_2_ Reduction Activity

**DOI:** 10.3390/molecules28010150

**Published:** 2022-12-24

**Authors:** Chenjiao Huang, Wenwen Bao, Senhe Huang, Bin Wang, Chenchen Wang, Sheng Han, Chenbao Lu, Feng Qiu

**Affiliations:** 1School of Chemical and Environmental Engineering, Shanghai Institute of Technology, 100 Haiquan Road, Shanghai 201418, China; 2The Meso-Entropy Matter Lab, State Key Laboratory of Metal Matrix Composites, Frontiers Science Center for Transformative Molecules, School of Chemistry and Chemical Engineering, Shanghai Jiao Tong University, 800 Dongchuan Road, Shanghai 200240, China

**Keywords:** Co(II) porphyrin, carbon dioxide reduction, push–pull effect, electrocatalysis, faradaic efficiency

## Abstract

Molecular electrocatalysts for electrochemical carbon dioxide (CO_2_) reduction has received more attention both by scientists and engineers, owing to their well-defined structure and tunable electronic property. Metal complexes via coordination with many π-conjugated ligands exhibit the unique electrocatalytic CO_2_ reduction performance. The symmetric electronic structure of this metal complex may play an important role in the CO_2_ reduction. In this work, two novel dimethoxy substituted asymmetric and cross-symmetric Co(II) porphyrin (PorCo) have been prepared as the model electrocatalyst for CO_2_ reduction. Owing to the electron donor effect of methoxy group, the intramolecular charge transfer of these push–pull type molecules facilitates the electron mobility. As electrocatalysts at −0.7 V vs. reversible hydrogen electrode (RHE), asymmetric methoxy-substituted Co(II) porphyrin shows the higher CO_2_-to-CO Faradaic efficiency (FE_CO_) of ~95 % and turnover frequency (TOF) of 2880 h^−1^ than those of control materials, due to its push–pull type electronic structure. The density functional theory (DFT) calculation further confirms that methoxy group could ready to decrease to energy level for formation *COOH, leading to high CO_2_ reduction performance. This work opens a novel path to the design of molecular catalysts for boosting electrocatalytic CO_2_ reduction.

## 1. Introduction

Recently, metal complexes consisting of transition metal ions with heteroatom-embedded organic molecules as ligands are emerging as good catalyst materials in wide electrochemical application, due to the existence of occupied d_z_ orbitals for the favorable catalytic CO_2_ reduction activity [1,2,3,4]. As the candidates of a molecular electrocatalyst, the metal complex with well-defined molecular structure could be ready to control by changing of various metal ions and organic ligands (e.g., dipyridine, terpyridine, dipyrromethane, porphyrin) with different π-conjugated systems, achieving tunable electrocatalytic performance [5,6,7]. Moreover, these metal complexes could be used as key building blocks for the preparation of organic porous polymers and single-atom carbon materials [8,9]. Different from other metal complexes, porphyrin possesses planar macrocyclic aromatics with extended π-electron conjugation, endowing them with some optical/electronic characteristics, like a broad photoabsorption wavelength, a narrow bandgap, and fast electron acceptors, among other properties [8,10,11,12,13,14]. Therefore, metal porphyrins have been proven as good candidates to be the electrocatalysts for CO_2_ reduction. So far, the research of porphyrin-based molecular electrocatalysts mostly focuses on the development of new porphyrin-based ligands for the improvement of CO_2_ reduction reactions (CO_2_RR).

To date, diverse analogues of porphyrins (including N-confused porphyrins [15], tetraaza [14], annulenes [16], and conjugated N_4_-macrocyclic ligand [17]) have been reported to coordinate with a large number of metal ions, exhibiting unique optoelectronic properties. In another way, modification of porphyrins with functional groups has been confirmed as a good strategy for improving their CO_2_RR performance [18,19,20]. For example, we have reported that the tertiary amine group could enhance CO production due to the enrichment of CO_2_ around molecules by amine groups [18]. Recently, some π-conjugated groups (like azulene, pyrene, etc.) were grafted onto the metal tetraphenylporphyrins to extend their conjugated system with lower bandgaps, resulting in the high electrocatalytic CO_2_ conversion [19,20]. Unfortunately, synthesis of these molecules often suffers from tedious synthetic steps with low reaction yields. Alternatively, side-chain engineering has been a versatile way to achieve functionalization of organic semiconductors [21,22,23,24,25]. For example, the push–pull structure prepared by side-chain engineering could enhance the electron delocalization for charge transfer, leading to the better electyrocatalysis [26,27]. Recently, Yaghi et al. reports that methoxy substituted Co(II) porphyrin-based covalent organic frameworks exhibit better electrocatalytic CO_2_ reduction, but are still far from satisfactory due to the low π-conjugation of imine bonds [22]. We also have found that methoxy substituent could efficiently tailor physical properties of conjugated polymers [28]. Owing to the commercial gain of methoxy-substituent aromatics, methoxy-functionalized metal porphyrins are easy to prepare and apply for electrochemical CO_2_RR. In addition, the topological structure of methoxy-functionalized metal porphyrins also is seldom exploited.

Herein, a novel kind of methoxy-functionalized Co(II) porphyrins were prepared via the conventional organic synthesis method. The chemical structures, optical/electronic properties of *as*-PorCo-OMe and *cs*-PorCo-OMe were well investigated. As electrocatalysts, these PorCo show the distinct electrocatalytic CO_2_ reduction. At −0.7 V versus. RHE, *as*-PorCo-OMe achieves a superior CO_2_RR performance, including FE_CO_ of 94.7% and TOF of 2880 h^−1^ at −0.7 V vs. RHE to the *cs*-PorCo-OMe. Furthermore, DFT also demonstrates that the methoxy substituent favors the push–pull effect on the porphyrin backbone, leading to the enhanced electrocatalytic CO_2_RR activity.

## 2. Results

### 2.1. Synthesis Description

The synthetic route to two methoxy-functionalized CoPor (asymmetric and cross-symmetric CoPor named as *as*-PorCo-OMe and *cs*-PorCo-OMe, respectively) are given in Figure 1a. The key intermediate of imine-containing dipyrromethane derivate (DMP-imine) was prepared from 2,6-dimethylbenzaldehyde using a three-step reaction in total yield of 57%, according to the reported work [29]. The 2,2′-((2,5-dimethoxyphenyl)methylene)bis(1*H*-pyrrole) (DmpMP) was synthesized by condensation reaction of 2,5-dimethoxybenzaldehyde with pyrrole in good yield of 82%. Then, *as*-PorCo-OMe was prepared by a one-pot reflux reaction of DmpMP, DMP-imine and cobalt acetate [Co(OAc)_2_] in ethanol for 18 hrs. The pure *as*-PorCo-OMe was purified by alumina column chromatography with PE and DCM (*v/v* = 8:2) as a crimson solid in the yield of 17%. For *cs*-PorCo-OMe, the 5,10,15,20-tetrakis(2,5-dimethoxyphenyl) porphyrin was firstly synthesized from 2,5-dimethoxybenzaldehyde and pyrrole in propionic acid for 24 h, and the crude purple solid was filtered and used without purification. After reaction with Co(OAc)_2_ in DMF, the *cs*-PorCo-OMe was obtained and purified by alumina column chromatography with PE and DCM (*v/v* = 6:4) as a crimson solid in yield of 15%. The detailed synthesis information and NMR spectra on these compounds is provided in Section 4 and Appendix A). The target molecular weight of *as*-PorCo-OMe and *cs*-PorCo-OMe are confirmed by mass spectrometry (MS). Figure 1b shows that the MS results are consistent with the predicted values of targeted compounds, suggesting the successful preparation of *as*-PorCo-OMe and *cs*-PorCo-OMe.

### 2.2. Structural Characterization

The structures of the as-prepared complexes were characterized by Fourier transform infrared (FTIR) and X-ray photoelectron spectroscopy (XPS). As shown in Figure 2a, the absorption bands between 1596 and 1445 cm^−1^ are attributed to the C = C vibration peaks of aromatic (like phenyl and pyrrole) groups, and absorption band at 1352 cm^−1^ is the C=N bond in the backbone of porphyrin [30], while the peak at 997 cm^−1^ is associated with the vibration of the Co-N bond [31]. These results demonstrate the successful preparation of Co(II) porphyrin derivates. The stretching vibration peaks of C-H is at 2965 and 2922 cm^−1^ for methyl and aromatic groups, respectively [18]. Moreover, the intensity of peak at 2965 cm^−1^ in *cs*-PorCo-OMe is stronger than that of *as*-PorCo-OMe, due to existence of eight methoxy group in *cs*-PorCo-OMe. The C-O bands in *as*-PorCo-OMe and *cs*-PorCo-OMe are found at 1211 and 1258 cm^−1^, respectively, suggesting the stronger conjugation effect of the methoxy bond in *cs*-PorCo-OMe [32,33]. The chemical states of elementals of these complexes have also been investigated. Appendix A show that elements of cobalt (Co), carbon (C), oxygen (O) and nitrogen (N) are displayed and both two complex show the similar high-resolution XPS results. In Figure 2b, the Co 2p high-resolution XPS spectra of *as*-PorCo-OMe exhibits two main peak at 779.1 and 794.8 eV, resulting from Co(II) atom with Co 2p^3/2^ and Co 2p^1/2^ binding energies, respectively [18]. For N 1s XPS spectra, these complexes exhibit two peaks at 397.8 and 401.1 eV, attributing to the pyrrolic N and Co-N structures, respectively (Figure 2c) [34]. In addition, the C 1s XPS spectra can be separated into three peaks at 284.1, 286.1 and 287.4 eV, indicating the bend energy of C-C/C=C, C-O and C=N, respectively (Figure 2d) [35]. These results demonstrate the accurate structure of Co(II) porphyrins with various methoxy substituents.

### 2.3. Electronic Structures

The photophysical properties of as-synthesized materials was investigated by ultraviolet and visible adsorption (UV–Vis) spectroscopy in dichloromethane (DCM) (Figure 3a). The Soret band of *as*-PorCo-OMe shows a strong absorbance at 398 nm, indicating the π-π* transition of porphyrin backbones, while its Q band is located between 498 and 564 nm indicating the n-π* transition from donor-acceptor structure [36]. Compared with that of *as*-PorCo-OMe, *cs*-PorCo-OMe has the enhanced push–pull effect, due to the increasing number of 2,5-dimethoxyphenyl groups, leading to the obvious red-shift phenomenon in the UV-Vis spectrum [37]. Furthermore, the board single peak of Q band suggests the symmetric structure of *cs*-PorCo-OMe [29]. On the basis of their UV-Vis results, the optical bandgap (E_g_) of *as*-PorCo-OMe and *cs*-PorCo-OMe can be calculated to be 2.17 and 2.18 eV, respectively, by using Tauc measurement (Figure 3b). The decrease of bandgap in these complexes manifests the donor effect of the methoxy group, but, the slight change is caused by steric effect of α-functionalized methoxy substituent.

The cyclic voltammetry (CV) measurement was exploited to characterize to the electronic structures of methoxy-substituted Co(II) porphrins. The CV curves, performed in argon (Ar)-saturated 0.1 M TBAPF_6_ DCM solution, are given in Figure 3c. The *as*-PorCo-OMe exhibits an irreversible one-electron reduction, while *cs*-PorCo-OMe has two successive reduction processes, indicating that the electron could be delocalized effectively over the molecular backbone, due to the symmetric structure of *cs*-PorCo-OMe [38]. The peak around −0.7~−0.9 V is the reduction reaction of Co(II)-to-Co(I) [39]. Based on the onset of first reduction potential, the LUMO energy level of *as*-PorCo-OMe and *cs*-PorCo-OMe is −3.39 and −3.63 eV, respectively. Following the equation:HOMO = LUMO − E_g_,(1)
the HOMO energy levels of *as*-PorCo-OMe and *cs*-PorCo-OMe are calculated as −5.57 and−5.80 eV, respectively (Figure 3d) [40].

### 2.4. DFT Calculation

To gain deep insight into the electronic and geometric structures, frontier orbitals of these complexes were performed by DFT calculations (Figure 4a). The LUMOs of *as*-PorCo-OMe and *cs*-PorCo-OMe mainly reside on the backbone of porphyrin, indicative of their similar LUMO energy level at −2.05, and −2.08 eV, respectively. For HOMO, the porphyrin core and partial 2,5-dimethoxybenzene are covered, demonstrating the donor effect of methoxy group in the substituents [40]. With the increasing number of substituents, the push–pull effect becomes stronger. The calculated energy levels of methoxy-substituted porphyrins are well agreement with the tested results from CVs, and the detailed information is provided in Table 1.

Based on pervious works, the Co(II) porphyrin has been approved as a good candidate to be the electrocatalyst for the CO_2_-to-CO reduction via a four-step reaction [41,42]. Thus, the DFT was carried out to investigate the reaction kinetics of the electrochemical CO_2_ reduction process with as-synthesized molecular catalysts (Appendix A). As shown in Figure 4b, the formation of *COOH is the rate-limiting step in CO_2_RR in this reaction energetics evolution. The free energy path of the conversion of CO_2_ to *COOH (Δ_*GCOOH_) requires 0.65 and 0.64 eV, respectively, for *as*-PorCo-OMe, and cs-PorCo-OMe, which is similar to that of 5,15-bis(2,6-dimethylphenyl) Co(II) porphyrin (DMP-CoPor). The methoxy substitution could provide the electron donor effect on a bit of enhancement of electrocatalytic activity.

### 2.5. Electrocatalytic CO_2_RR

The electrocatalytic CO_2_RR performance of methoxy-substituted Co(II) porphyrins were evaluated in a the 0.5 M KHCO_3_ electrolyte using an H-type three-electrode cell with Nafion-117 as separator. All potentials are applied to the reversible hydrogen electrode (RHE) [43]. The electrocatalytic activity of methoxy-substituted Co(II) porphyrins in the Ar- and CO_2_-saturated electrolyte was studied by the linear sweep voltammetry measurement. Appendix A illustrates that the current densities of these molecules is higher in CO_2_ atmosphere than that in the Ar-saturated condition, suggesting the presence of electrocatalytic activity of the Co(II) porphyrin core [44]. As shown in Figure 5a, both of *as*-PorCo-OMe and *cs*-PorCo-OMe generate the increasing current densities with the increase of potential from −0.4 to −1.0 V versus RHE. Compared with *cs*-PorCo-OMe, the *as*-PorCo-OMe shows higher electrocatalytic activity. This result may result from the lower steric hindance effect of *as*-PorCo-OMe than that of *cs*-PorCo-OMe, leading to the fast electron transfer from carbon nanotubes to catalysts for enhanced electrochemical CO_2_RR [29].

The CO_2_RR products were tested by the online gas chromatography (GC) and off-line NMR techniques (Appendix A), which confirms that only CO and H_2_ were found during the reduction reaction, suggesting the high selectivity of methoxy-substituted Co(II) porphyrins. The CO Faraday efficiencies (FE_CO_) of two complexes are given in Figure 5b. As expected, the FE_CO_ of *as*-PorCo-OMe reaches as high as 94.7%, which is much larger than those of DMP-CoPor (85.5%) [18], suggesting that the electron donor of methoxy substitution has the positive influence for electrocatalytic CO_2_ reduction by push–pull effect. Moreover, FE_CO_ of *as*-PorCo-OMe also is better than that of *cs*-PorCo-OMe (84.5%), as well as reported PorCo-TPP (91%) [39], PorCo-MOF (76%) [45] and Co proto-porphyrin (40%) [46]. Correspondingly, the partial current densities of methoxy-substituted CoPors for CO production increase with the increase of potentials. As the example of at −0.7 V, the specific current density of *as*-PorCo-OMe is over two times higher than that of *cs*-PorCo-OMe. The catalytic activities of these methoxy-substituted Co(II) porphyrins was comprehensively evaluated by the index of turnover frequency (TOF). In Figure 5d, the TOF values of two molecules gradually increased from the potential from −0.4 and −1.0 V vs. RHE, owing to the increase of current density at high potential. Compared with *cs*-PorCo-OMe, *as*-PorCo-OMe exhibits better TOF performance in the whole potentials, indicative its good electrochemical activity for CO_2_RR application. Furthermore, the TOF of *as*-PorCo-OMe (2880 h^−1^ at −0.7 V vs. RHE) also is superior to many reported state-of-the-art porphyrin-based electrocatalysts [10,18,29,45,46,47,48,49,50]. These results demonstrate the efficient electron and proton transfer kinetics for the push–pull type *as*-PorCo-OMe with weak steric hindrance.

The Tafel slope represents a reaction kinetic of rate determining steps involved in electrocatalysis, which can be calculated from the polarization curves [51]. In Figure 6a, the *as*-PorCo-OMe shows the Tafel value of 145 mV dec^−1^, which is smaller than that of *cs*-PorCo-OMe (200 mV dec^−1^), indicating that *as*-PorCo-OMe has the higher catalytic activity of *COOH formation in CO_2_ reduction reaction via electron/proton transfer [52,53]. To evaluate the electrochemical behavior of as-prepared complexes in electrocatalytic CO_2_RR, electrochemical impedance spectroscopy (EIS) was carried out [54]. The charge transfer resistance (R_ct_) derived from the Nyquist plot exhibits that the resistance of 33.25 Ω for *as*-PorCo-OMe is lower than that of *cs*-PorCo-OMe (38.69 Ω) (Figure 6b), demonstrating the superior electron transfer ability of *as*-PorCo-OMe. Furthermore, the electrochemical capacitances from CV between −0.26 and −0.16 eV vs. RHE show that *as*-PorCo-OMe provides the higher electrochemical active surface area (Figure 6c and Appendix A), benefiting from its asymmetric push–pull structure and low steric hindrance effect. Thus, *as*-PorCo-OMe has been approved as the good catalyst for electrocatalytic CO_2_RR application. The durability performance of *as*-PorCo-OMe was investigated at −0.7 V vs. RHE (potential for best FE_CO_) (Figure 4d). After testing for 12 hr, the FE_CO_ of *as*-PorCo-OMe remains over 93% and its current density has a low loss, and Appendix A shows that Co 2p and N1s XPS spectra have a neglect binding energy change after the cycle experiment, demonstrating that such asymmetric Co(II) porphyrin exhibits a good electrochemical stability during long-term working.

## 3. Conclusions

In summary, a novel kind of push–pull type Co(II) porphyrins with methoxy substitutions have been prepared efficiently by using 2,5-dimethoxybenzaldehyde as starting material. The structures of these methoxy-substituted molecules have been confirmed by various measurement like MALDI-TOF MS, FTIR and XPS spectroscopy. Compared with that of *as*-PorCo-OMe, *cs*-PorCo-OMe shows the slight red-shift absorption properties and low bandgap, due to the limited donor effect of methoxy substitution in these structures. Such as-prepared Co(II) porphyrins bearing electrocatalytic active site of cobalt ion would be applied as electrocatalysts for CO_2_RR. In a CO_2_-saturated KHCO_3_ aqueous solution, *as*-PorCo-OMe exhibits the better electrochemical CO_2_-to-CO performance including FE_CO_ of 94.7% and TOF of 2880 h^−1^ at −0.7 V vs. RHE than those of *cs*-PorCo-OMe and reported DMP-CoPor, which is almost in agreement with that of DFT calculation. Therefore, this work provides a new molecular engineering strategy for boosting electrocatalytic CO_2_RR via methoxy functionalization.

## 4. Materials and Methods

### 4.1. Materials

Pyrrole, 2,5-dimethoxybenzaldehyde, BF_3_•Et_2_O, propionic acid, and cobalt acetate were purchased from Adamas. The DMP-imine has been prepared according to previous work. Organic solvents including chloroform (CHCl_3_), dichloromethane (CH_2_Cl_2_), petroleum ether (PE), dimethyl Formamide (DMF), ethyl acetate (EA), ethanol (EtOH) and all other materials were used without further purification.

### 4.2. Synthesis Procedures

Synthesis of 2,2′-((2,5-dimethoxyphenyl)methylene)bis(1H-pyrrole) (DpmPM). In a 250 mL flask, 2,5-dimethoxybenzaldehyde (4.98 g, 30.0 mmol) and pyrrole (145 mL, 2.10 mol) was stirred under an N_2_ atmosphere for 30 min. BF_3_·OEt_2_ (3.56 g, 25.0 mmol) was added into the solution, and kept stirring at room temperature for 2 h. Then, NaOH (9.00 g, 225 mmol) was added for another 1 h. The crude product was received from the filtrate under reduced pressure after filtering the mixture of reaction. The product was purified by column chromatography with EtOAc and PE (*v:v* = 10:90) to afford a pale yellow solid product (6.8 g, 80%). ^1^H NMR (CDCl_3_, 500 MHz): δ (ppm) = 3.71 (d, 6H, J = 4.25 Hz, OCH_3_), 5.75 (s, 1H, CHC3), 5.92 (s, 2H, Py-H), 6.13 (q, 2H, J = 8.08 Hz, Py H), 6.66 (q, 2H, J = 7.60 Hz, Py H), 6.71 (d, 1H, J = 2.77 Hz, Ar-H), 6.76 (m, 1H, Ar-H), 6.84 (d, 1H, J = 7.08 Hz, Ar-H), 8.15 (s, 8H, H-pyrrole).

Synthesis of *as*-PorCo-OMe. In a 250 mL flask, DMP-imine (2.00 g, 5.15 mmol), DpmPM (1.48 g, 5.27 mmol) and Co(OAc)_2_ (9.44 g, 51.5 mmol) were mixed in ethanol (250 mL) under an N_2_ atmosphere for 30 min. Then, the solution was stirred at 80 °C for 24 h. After the reaction, a dark-purple solid was collected by vacuum, and was purified by alumina column chromatography (PE/DCM = 8:2) to obtain as-PorCo-OMe (445 mg, 14%).

Synthesis of *cs*-Por-OMe. Pyrrole (400 mg, 5.96 mmol) and 2,5-dimethoxybenzaldehyde (1002 mg, 6.04 mmol) were dissolved in propionic acid (200 mL). The solution was heated to 110 °C under an N_2_ atmosphere. After stirring for 24 h, the crude product was obtained via precipitation in the methanol. The pure purple solid was obtianed by washing with methanol until it was a transparent color (612 mg, 12%). ^1^H NMR (CDCl_3_, 500 MHz): δ (ppm) = −2.66 (s, 2H, NH), 3.51 (m, 12H, OCH_3_), 3.91 (m, 12H, OCH_3_), 7.23–7.31 (m, 8H, Ar-H), 7.55–7.66 (m, 4H, Ar-H), 8.78 (s, 8H, H-pyrrole).

Synthesis of *cs*-PorCo-OMe. The obtained *cs*-Por-OMe (300 mg, 0.35 mmol) and Co(OAc)_2_ (800 mg, 4.52 mmol) was dissolved in DMF (30 mL). The solution was heated to 100 °C for 8 h under an N_2_ atmosephere. After reaction, the solvent was removed and the solid was precipitated in the methanol and purified by a silica gel column chromatography (PE/DCM = 6:4) to collect *cs*-PorCo-OMe (304 mg, 95%).

### 4.3. Characterizations

NMR spectra were obtained from a Bruker Avance III 500 MHz spectrometer using CDCl_3_ as solvents. MALDI–TOF mass spectrometry was recorded on autoflex speed^TM^ TOF Mass Spectrometer. FTIR spectra were performed on Perkin Elmer Spectrum 100 spectrometer with KBr. XPS spectra were measured with a PHI 5000C ESCA System using C 1s (284.8 eV) as reference. UV–Vis spectra were recorded on a Lambda 950 spectrophotometer. CV tests were performed using 0.1 M TBAPF_6_ DCM solution as an electrolyte with the CH CHI 660E instrument.

### 4.4. Electrode Preparation

Firstly, catalysts (1 mg) were dispersed well in the commercial CNTs (9 mg) (Appendix A), then Nafion solution (2 mL, 0.5 wt. %) was added and stirred for 12 h. A quantity of 100 µL of mixed ink was dropped on carbon paper (surface: 1 cm^2^) until dry to achieve the working electrode with catalyst loading of 0.05 mg cm^−2^.

## Figures and Tables

**Figure 1 molecules-28-00150-f001:**
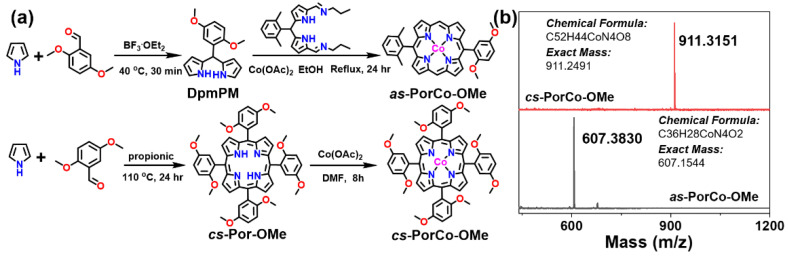
(**a**) synthetic route to the *as*-PorCo-OMe and *cs*-PorCo-OMe; (**b**) MALDI-TOF mass spectra of *as*-PorCo-OMe and *cs*-PorCo-OMe.

**Figure 2 molecules-28-00150-f002:**
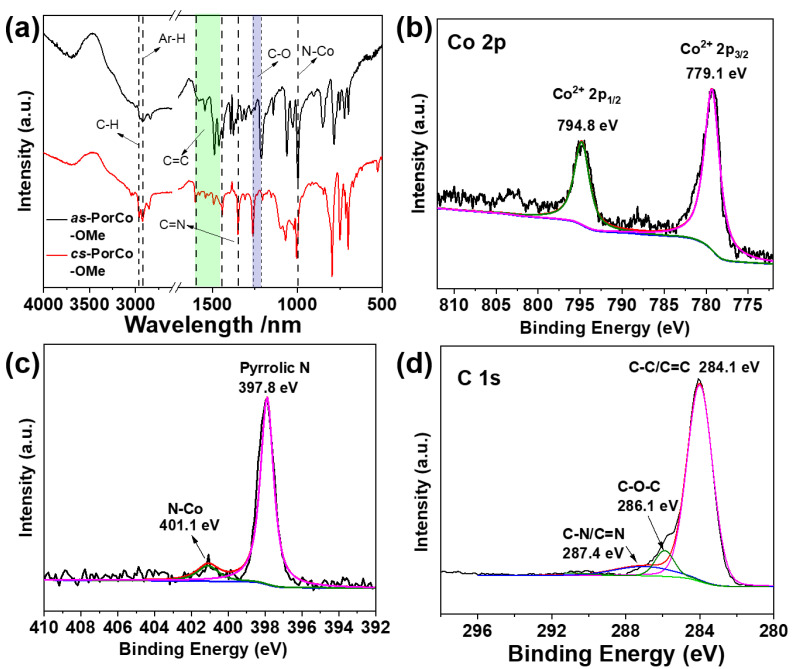
(**a**) FTIR spectra of *as*-PorCo-OMe and *cs*-PorCo-OMe; High resolution Co 2p (**b**), N 1s (**c**) and C 1s (**d**) XPS spectra in *as*-PorCo-OMe.

**Figure 3 molecules-28-00150-f003:**
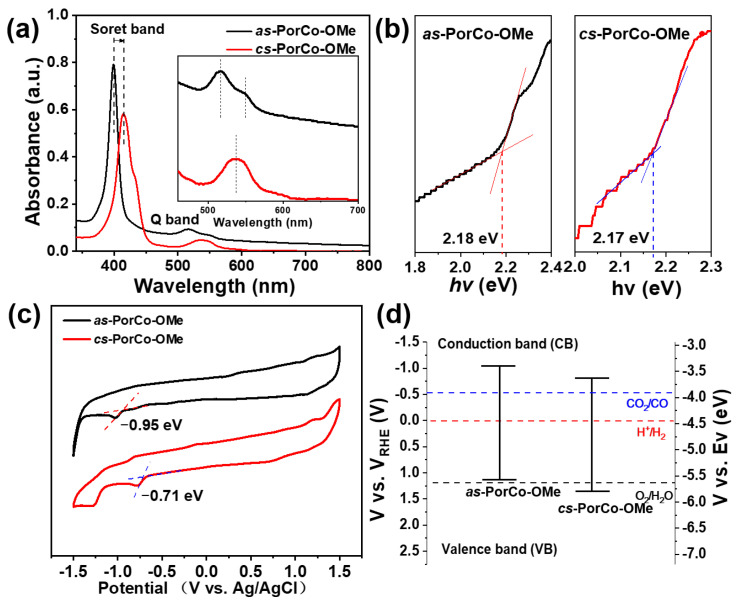
(**a**) UV-vis absorption spectra of *as*-PorCo-OMe and *cs*-PorCo-OMe; (**b**) bandgap of *as*-PorCo-OMe and *cs*-PorCo-OMe calculated by Tauc method; (**c**) cyclic voltammetry curves of *as*-PorCo-OMe and *cs*-PorCo-OMe; (**d**) Band structure diagram for *as*-PorCo-OMe and *cs*-PorCo-OMe.

**Figure 4 molecules-28-00150-f004:**
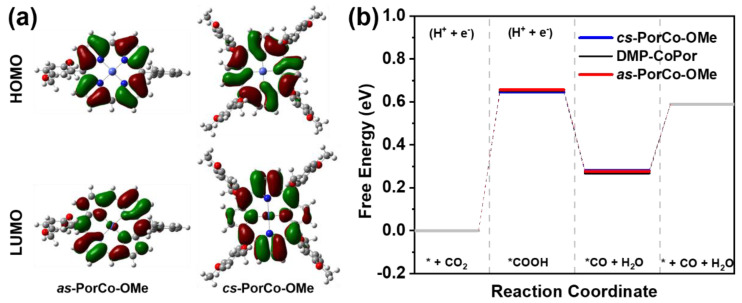
(**a**) Calculated HOMO and LUMO levels of of *as*-PorCo-OMe and *cs*-PorCo-OMe; (**b**) Free energy of *as*-PorCo-OMe and *cs*-PorCo-OMe in different CO_2_RR steps.

**Figure 5 molecules-28-00150-f005:**
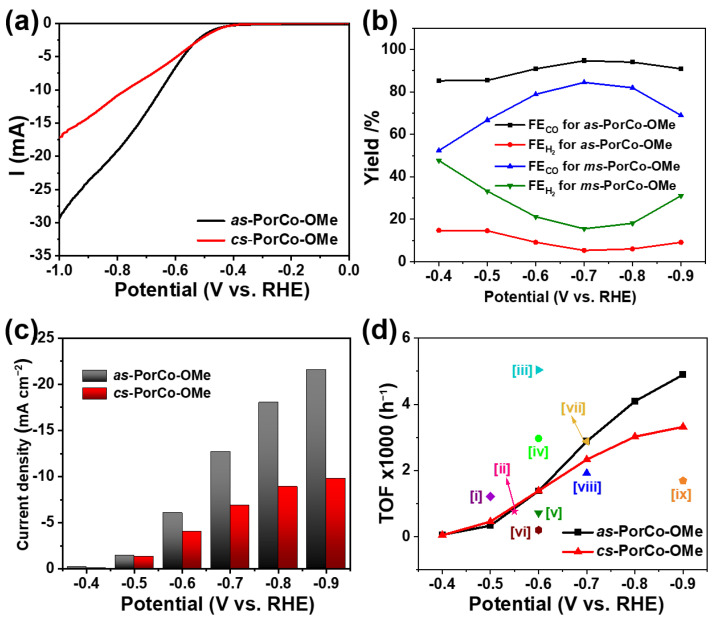
(**a**) LSV curves of *as*-PorCo-OMe and *cs*-PorCo-OMe in CO_2_-saturated 0.5 M KHCO_3_ electrolyte (scan rate: 5 mV s^−1^); (**b**) FE_CO_ and FE_H2_ of of *as*-PorCo-OMe and *cs*-PorCo-OMe at various specific potentials; (**c**) CO partial current densities at various specific potentials; (**d**) Comparison of TOF of as-prepared complex with various porphyrin-based electrocatalysts of [i] PorFe-MOF [47], [ii] COF-367-PorCo (1%) [10], [iii] PorCo/cationic POP [48], [iv] *as*-PorCo [29], [v] CoTMPP [49], [vi] PorCo-MOF [45], [vii] Co protoporphyrin [46], [viii] DMP-CoPor [18], [ix] PorNi-CTF [50].

**Figure 6 molecules-28-00150-f006:**
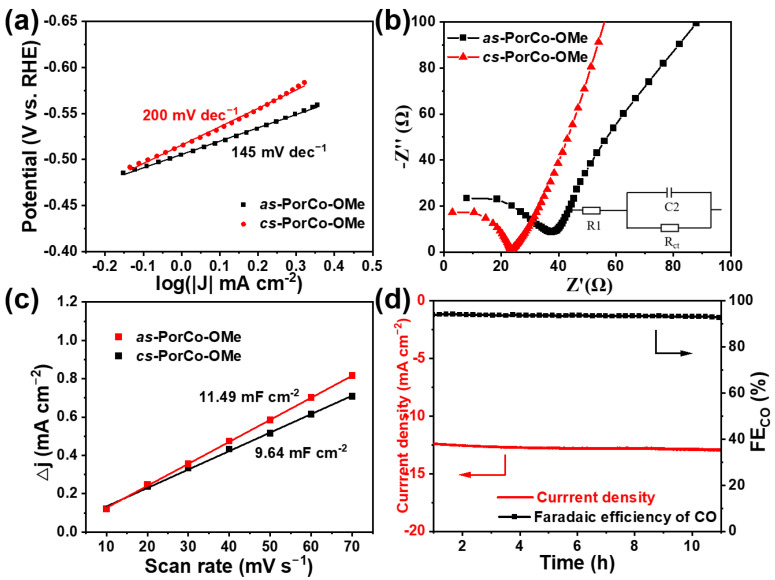
(**a**) Tafel slopes and (**b**) Nyquist plots of *as*-PorCo-OMe and *cs*-PorCo-OMe; (**c**) capacitive current as a as a function of scan rate for *as*-PorCo-OMe and *cs*-PorCo-OMe; (**d**) long-term stability of *as*-PorCo-OMe at −0.7 V vs. RHE for 12 h.

**Table 1 molecules-28-00150-t001:** Electrochemical data of *as*-PorCo-OMe and *cs*-PorCo-OMe.

Entry 1	E_cv.red_ (V) ^[a]^	E_cv,LUMO_ (eV) ^[b]^	E_cv.HOMO_ (eV) ^[c]^	E_opt.gap_ (eV) ^[d]^	E_DFT,LOMO_ (eV) ^[e]^	E_DFT,HOMO_ (eV) ^[e]^	E_DFT,gap_ (eV) ^[e]^
*as*-PorCo-OMe	−0.95	−3.39	−5.57	2.18	−2.05	−5.13	3.08
*cs*-PorCo-OMe	−0.71	−3.63	−5.80	2.17	−2.08	−5.12	3.04

[a] E_cv.red_ is the onset value of reduction potential. [b] For all molecules, *E*_ferrocene(FOC)_ = 0.46 V vs. Ag/AgCl; calculated LUMO levels based on the following equation: LUMO = −[*E*_cv.red_ − *E*_FOC_] − 4.8 eV. [c] HOMO = LUMO-E_opt.gap_. [d] Bandgaps determined from the UV/Vis absorption spectra using the Tauc method. [e] Calculated HOMO and LUMO levels and bandgap based on DFT simulation.

## Data Availability

The data presented in this study are available on request from the corresponding author.

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
