# Peer review of "Asymmetric Push–Pull Type Co(II) Porphyrin for Enhanced Electrocatalytic CO2 Reduction Activity"

_molecules, 2022, doi:10.3390/molecules28010150_

Round 1
Reviewer 1 Report
In this work, the authors have prepared a methoxy-substituted Co(II) porphyrin catalyst and studied its electrocatalytic performances in CO2 reduction. This manuscript is well prepared. I recommend its publication in Molecules.
1. For better understanding, the authors should give a brief explanation of the push-pull effect.
2. XPS profile needs to be carefully fitting the peaks in Figure 2 and S6.
3. In Figure 4b, the author should inset a magnification figure to clearly show the difference of three molecule catalysts.
4. In CO2RR test, the competitive HER should be mentioned.
5. The stability of catalyst upon electrocatalytic CO2 reduction should be discussed.
Reviewer 2 Report
This is an interesting study about asymmetric Push-pull type Co(II) porphyrin for efficient CO2 reduction to CO. The background about the application of Push-pull type porphyrin molecules should be stated in the introduction which is vital to the readers.
The author mentioned that cs-PorCo-OMe has enhanced push-pull effect, red-shifted UV absorption, and decreased optical bandgap compared with that of as-PorCo-OMe, which is consistent with the DFT simulation result. The former also displays more positive onset potential, hinting more efficient delocalization of electron. However, as-PorCo-OMe catalyst exhibits higher FEco. The author attributes it to the different steric hindrance. In my opinion, the catalyst dosage may affect the product efficiency which should be taken into consideration.
Reviewer 3 Report
This paper synthesized two Co(II) porphyrin (PorCo) complexes and measured the electrocatalytic performance for CO2 reduction. In the tow complexes, asymmetric meth-oxy-substituted Co(II) porphyrin shows the high CO2-to-CO Faradaic efficiency (FECO) of ~95 % and turnover frequency (TOF) of 2880 h-1, due to its push-pull type electronic structure. Density functional theory (DFT) calculation further confirms that methoxy group could decrease the energy level for the formation of *COOH, leading to high CO2 reduction electrocatalytic performance. However, there are some questions need to be resolved:
1, How to explain the influence of the structure of as-PorCo-OMe and cs-PorCo-OMe on the catalytic activit?
2, From the theoretical calculation, the Gibbs free energy of as-PorCo-OMe and cs-PorCo-OMe are close, while the difference of Faraday efficiency and current density is so large. Please give a detailed explanation.
3, Acetylene black is usually used for electrode preparation. Why is CNT used here? How does CNT combine with catalyst molecules? How does the loading position affect the catalytic performance?
4, For heterogeneous catalysis of single molecule catalyst, the loading condition can largely affect the catalytic performance. It is suggested to well characterize the catalyst loading on CNT before considering the electrocatalytic performance.
5, The Co(II) porphyrin has been approved a good candidate as the electrocatalyst for CO2-to-CO reduction, It is suggested to quote some literatures on electrocatalysis of porphyrin based MOF (ACS Appl. Mater. Interfaces, 2021, 13, 54959, Nano Energy 2020, 67, 104233, Nat. Commun. 2018, 9, 4466)
Round 2
Reviewer 2 Report
Accept.